# Locating and Dating Land Cover Change Events in the Renosterveld, a Critically Endangered Shrubland Ecosystem

**Glenn R. Moncrieff** [1,2]

1 Fynbos Node, South African Environmental Observation Network, Private Bag X7, Rhodes Drive, Claremont 7735, South Africa; glenn@saeon.ac.za
2 Centre for Statistics in Ecology, Environment and Conservation, Department of Statistical Sciences, University of Cape Town, Private Bag X3, Rondebosch 7701, South Africa

**Abstract:** Land cover change is the leading cause of global biodiversity decline. New satellite platforms allow for monitoring of habitats in increasingly fine detail, but most applications have been limited to forested ecosystems. I demonstrate the potential for detailed mapping and accurate dating of land cover change events in a highly biodiverse, Critically Endangered, shrubland ecosystem—the Renosterveld of South Africa. Using supervised classification of Sentinel 2 data, and subsequent manual verification with very high resolution imagery, I locate all conversion of Renosterveld to non-natural land cover between 2016 and 2020. Land cover change events are further assigned dates using high temporal frequency data from Planet labs. A total area of 478.6 hectares of Renosterveld loss was observed over this period, accounting for 0.72% of the remaining natural vegetation in the region. In total, 50% of change events were dated to within two weeks of their actual occurrence, and 87% to within two months. The Renosterveld loss identified here is almost entirely attributable to conversion of natural vegetation to cropland through ploughing. Change often preceded the planting and harvesting seasons of rainfed annual grains. These results show the potential for new satellite platforms to accurately map land cover change in non-forest ecosystems, and detect change within days of its occurrence. There is potential to use this and similar datasets to automate the process of change detection and monitor change continuously.

**Keywords:** land cover monitoring; change detection; threatened ecosystems; Sentinel 2; planet labs; Cape Floristic region

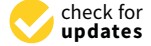

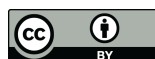

## 1. Introduction

Land cover change is the leading global cause of species extinctions [1]. Humans have significantly altered 75% of the earth's surface, with over one third of the terrestrial land surface currently being used for cropping or animal husbandry [2,3]. Hotspots of biodiversity have, on average, been more altered and degraded than other areas despite their importance [1]. The Cape Floristic Region (CFR) of South Africa is one such global biodiversity hotspot [4]. The CFR covers 90,000 km$^2$ and is home to around 9400 plant species, 68% of which are endemic [5]. This incredible plant species richness makes the CFR the most floristically diverse region of the world outside of the tropics [6]. The CFR is characterised by very high fine-scale habitat partitioning among species (beta diversity) and turnover of species in space (gamma diversity) [7]. Species within the CFR are thus inherently vulnerable to extinction through habitat loss, as many have narrow environmental tolerances and small range sizes [8,9].

Within the CFR, the vegetation characteristic of the lowlands is known as Renosterveld. Renosterveld typically occurs on fertile, shale-derived soils and has a significant grass component. Due to the association of Renosterveld with soils and topography suitable for agriculture, an estimated 90–94% of its original extent has been completely transformed into intensively cultivated dryland crops: predominantly wheat, canola and barley [10–12].

Sheep are grazed on fallow fields and rotational crops as well as Renosterveld patches, further degrading the few remaining intact fragments. As a result of this historical land cover change and habitat degradation, the Renosterveld contains perhaps the highest concentration of threatened plant species of any continental region globally [13–16]. Within the CFR as a whole, 1893 plant species are threatened with extinction, with many of these restricted to Renosterveld [11]. Of the 29 Renosterveld vegetation types described, 17 are listed as as either Critically Endangered, Endangered or Vulnerable according to South Africa's National Environmental Management: Biodiversity Act No 10 of 2004 (NEMBA). This act is intended to safeguard these ecosystems, ensuring formal protection and mandating restrictions on any loss of natural habitat. Further protection is given by the Conservation of Agricultural Resources Act, 1983 (Act No. 43 of 1983) (CARA). This act precludes the cultivation of virgin soil, defined as soil undisturbed for over 10 years, without prior authorization.

Despite this legal protection, and concerted conservation efforts (e.g., the Cape Action for People and the Environment programme (CAPE) and the Overberg Renosterveld Conservation Trust), land cover transformation within the Renosterveld continues unabated [17,18]. Recent estimates suggest that up to 1% of what remains of the Renosterveld has being lost each year between 1990 and 2014, with extermination expected to be completed in less than 75 years if these rates continue [11]. It was thought that historical rates of vegetation loss were unlikely to be maintained as the fragments of Renosterveld which do remain occur within areas that are less suitable for agriculture, with steep slopes or rocky soils [19]. However, improving technology and deteriorating economic conditions can render areas previously thought to be of minimal agricultural value vulnerable [10,19,20]. In order to prevent the loss of the last remaining fragments of Renosterveld, conservation management agencies and environmental law enforcement require timely and accurate information on the location and timing of land cover transformation. This information can help identify the proximal and ultimate drivers of vegetation loss and identify areas that may be susceptible to future change. In recent years new algorithms and an increase in the availability of remotely sensed data have improved the temporal and spatial resolution at which land cover change can be monitored [21–24]. These improvements have been translated into platforms that are in use by practitioners to aid monitoring of land cover change globally (e.g., Global Forest Watch).

Continuous land cover change monitoring typically uses an unsupervised approach to detecting change by examining remotely sensed time series of vegetation activity from sensors such as Landsat or MODIS [21–24]. Change is identified when vegetation activity deviates from expectations derived by fitting linear and seasonal trends. This approach has been used to great success in forest ecosystems [25,26]. These systems are well suited for this application, as land cover change involves a very large shift in vegetation—from closed canopy forest to bare ground—that is comparatively easy to detect in the spectral information returned from satellites. Open-canopied and low tree cover ecosystems, such as grasslands and shrublands, have received less attention, presumably because land cover change in these systems is harder to detect [27]. Vegetation activity is overall much lower than in forests, with a high signal to noise ratio [28]. However, these ecosystems are the dominant land cover type in sub-Saharan Africa, make up >40% of the global total ecosystem organic carbon and contain a substantial proportion of the world's biological diversity [29]. Existing land cover change products for South Africa (e.g., [30]) summarize change over multi-year or multi-decadal time periods, and are not updated frequently enough to inform management interventions that may address the underlying drivers of change. Nor can they be used for enforcement and prosecution of illegal transformation as they lack the ability to accurately date the occurrence of an event. There is therefore an urgent need to assess the potential for continuous land cover change monitoring in non-forest ecosystems and improve the accuracy of available methods.

Here I use multi-temporal high and very-high resolution remote sensing imagery to map and accurately date land cover change events over a 4 year period from 2016 to 2020 in

highly biodiverse, highly threatened, shrubland habitats within the Renosterveld of South Africa. I outline a workflow using freely available and open source tools to automatically identify areas of potential change and confirm change events. I attempt to assign a date to each land cover change event and describe the feasibility thereof. I describe the spatial and temporal patterns of land cover change in the Renosterveld in relation to the drivers of change. The resulting dataset of located and dated land cover change events provides training data for algorithm development to accelerate the advancement of land cover change monitoring in the Renosterveld and similar ecosystems.

## 2. Materials and Methods

This study is limited to the lowlands Renosterveld occurring within the boundaries of the Overberg district municipality, an area of 12,241 km$^2$ (Figure 1). The Overberg region was selected due to the availability of recently compiled data on the extent of remaining natural land cover, familiarity with the region and the presence of local conservation partners. Five Renosterveld vegetation types occur within the Overberg: Eastern Rûens Shale Renosterveld, Central Rûens Shale Renosterveld, Western Rûens Shale Renosterveld, Rûens Silcrete Renosterveld and Breede Shale Renosterveld [31]. All these ecosystems are listed either as Critically Endangered or Endangered under NEMBA. Together they form the natural land cover of 4711 km$^2$ of the Overberg. The map of remaining natural vegetation in the Overberg is based on the combination of data on agricultural field boundaries with 2005 SPOT imagery to create a map of areas classified as transformed, degraded or natural at 1:50,000 scale [18]. This analysis identified 656 km$^2$ of the original extent of Renosterveld vegetation in the Overberg remaining as either degraded or natural. The analysis of land cover change presented here is limited to this area. A multi-step process was undertaken to (1) identify areas where Renosterveld was potentially lost between 2016 and 2020, (2) validate each site of potential loss to create an accurately digitized map of loss for this time period, and (3) for each location where loss was observed, assign a date on which this loss occurred. Figure 2 provides an overview of this method which is explained in detail below.

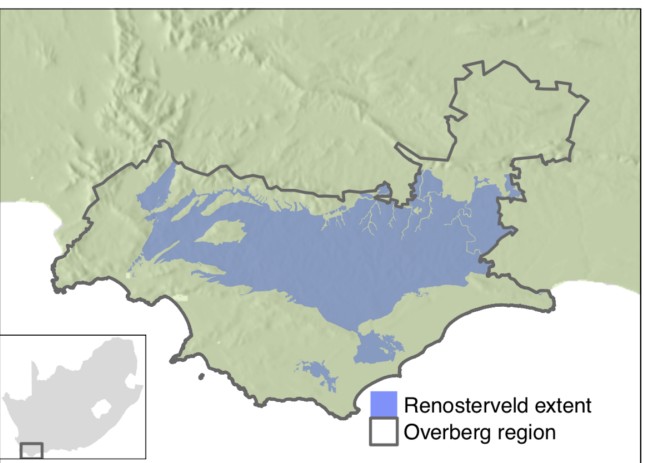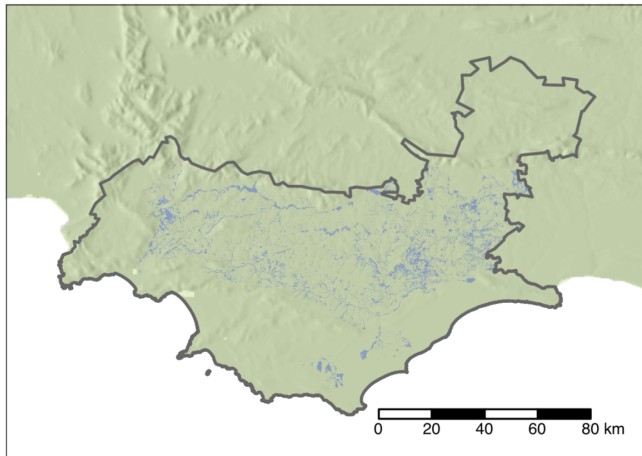

**Figure 1.** The Overberg district in South Africa. The **left panel** shows the natural extent of lowlands Renosterveld in the district and the **right panel** the remaining natural and degraded fragments.

### 2.1. Identifying Potential Change

Machine learning classification of Sentinel 2 imagery was used to identify areas of potential Renosterveld loss between 2016 and 2020, supervised using a manually created set of training data points from across the full extent of the study area. The year 2016 was chosen as the starting year for monitoring as it is the first full year for which both Sentinel 2 and PlanetScope data are extensively available. Sites with stable Renosterveld (no loss

between February 2016 and January 2020), stable agriculture (cultivated by February 2016) and Renosterveld loss (loss of Renosterveld between February 2016 and January 2020) were identified using very high resolution Google Earth imagery and 50 cm resolution aerial photography [32]. Any area under cultivation by February 2016 was considered stable agriculture, as it is assumed impossible for agriculture to revert to natural land cover over this time frame [33–35]. Points were chosen to cover a variety of agricultural land use types using a regional map of crop types [36]. Points for stable Renosterveld were located in areas with apparently intact Renosterveld in January 2020. It is impossible to confirm that sites were pristine, primary Renosterveld from remotely sensed imagery alone. However, the presence of shrubs and heterogeneous plant composition characteristic of Renosterveld stands in stark contrast to the monocultures or bare soil characteristic of agricultural land, and can thus be discerned with confidence by a trained eye. The reliability of Renosterveld identification from aerial imagery alone was confirmed by multiple ad hoc field excursions. Locations of loss of Renosterveld between February 2016 and January 2020 were created from identifying areas where the presence of Renosterveld could be confirmed in February 2016 and the presence of non-natural land cover could be confirmed in January 2020. The location of sites where change was suspected to have occurred between these dates was provided by local conservation partners. The final dataset of labelled sites consists of 2177 sites for stable Renosterveld, 2336 sites for stable agriculture, and 909 site of Renosterveld loss. QGIS 3.10 and Google Earth were used for image interpretation and digitization.

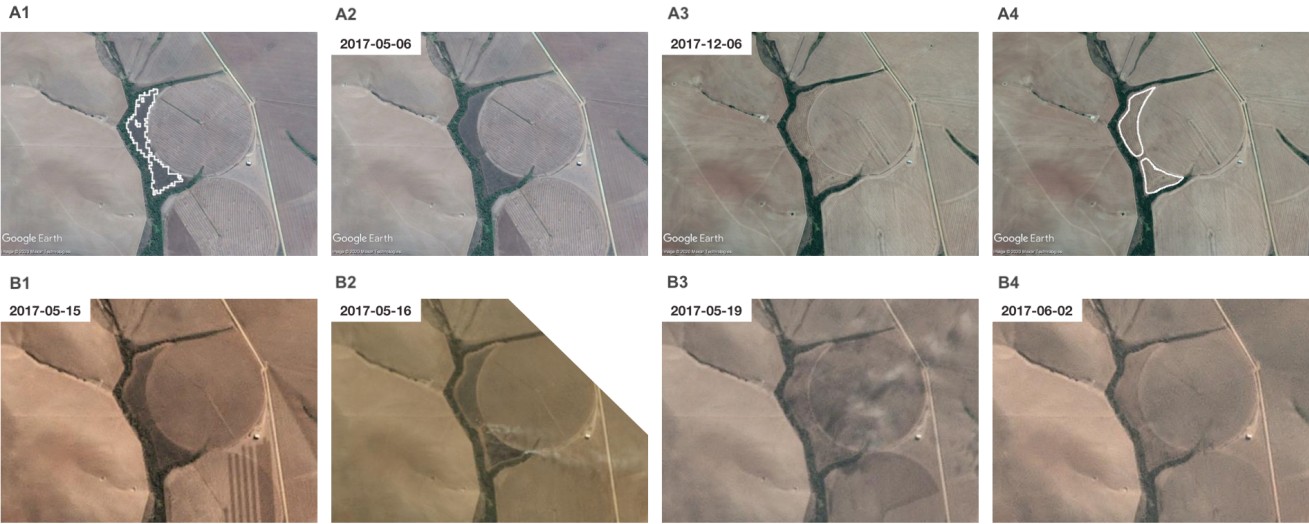

**Figure 2.** Outline of the methodology used for (**A**) accurate mapping of Renosterveld loss and (**B**) obtaining dates for the occurrence of change events. The output from a random forest classification outlines an area of potential Renosterveld loss (**A1**). Very-high resolution Google Earth imagery confirms the presence of Renosterveld on 6 May 2017 (**A2**) and its absence on 6 December 2017 (**A3**). With the help of very-high resolution imagery an accurate outline of the areas lost is obtained (**A4**). The time-series of PlanetScope imagery confirms 15 May 2017 as the last day that the presence of Renosterveld can be confirmed (**B1**). Vegetation removal is ongoing on 16 May 2017, suggested by the presence of smoke (**B2**). By 19 May 2017, no Renosterveld remains (**B3**), and this is confirmed by later imagery from 2 June 2017 (**B4**).

Input features were derived from Sentinel 2 L1C data from 6 February 2016 and 11 January 2020. Top-of-atmosphere Sentinel 2 data were used as surface reflectance maps and are not globally available for the entire Sentinel 2 record on Google Earth Engine. The entire study region falls within a single Sentinel 2 track (sensing orbit number 121) and thus images from a single date could be used for the beginning and end of the monitoring period, ensuring relatively homogeneous atmospheric conditions over the region. Dates were selected based on the availability of favourable atmospheric conditions and the absence of clouds, while mid- to late-summer was preferred due to the strong visual contrast

between fallow fields and Renosterveld at this time of year. Three spectral indices were calculated for each image: normalised difference vegetation index (NDVI), normalised difference red-edge index (NDRE), and the enhanced vegetation index (EVI). In addition to these, a single measure of spectral distance between these two images for each pixel was calculated using the spectral angle mapper algorithm [37]. Combining this with the calculated VI's and bands 2, 3, 4, 5, 6, 7, 8, 8A, 11 and 12 from each image results in a total of 27 input features. These features along with the labelled points were used to train a Random Forest classifier. Alternate classifiers (SVM, CART) were experimented with, but did not exceed the performance of Random Forests. The labelled data were split into five sections based on location, and spatial k-fold cross validation was used to assess performance [38]. The trained model was then used for prediction over the entire study region. The output of this prediction was filtered to retain only predictions of Renosterveld loss and masked to areas mapped as degraded or natural in 2008. Only areas of change larger than 0.1 ha or roughly ten contiguous pixels were retained for subsequent investigation. Google Earth Engine was used for data preparation, model fitting and evaluation [39].

### 2.2. Precisely Locating Change

Each of these potential land cover change events was then manually validated using very-high resolution Google Earth imagery, 50 cm resolution aerial photography [32] and high resolution (3–5m) PlanetScope imagery from Planet labs [40]. Every contiguous output parcel was checked to ensure that (1) landcover was indeed non-natural/agriculture at the end of the monitoring period, (2) intact Renosterveld was present at the start of the monitoring period, (3) land was uncultivated at least 10 years prior to the start of the monitoring period and (4) that the change took place between February 2016 and January 2020. If these four criteria were met, the accuracy of the output outline was improved by manually digitising by the area of Renosterveld lost at 1:50,000 scale. Each of these output Renosterveld change events was reviewed and confirmed by a second independent assessor and a local vegetation expert.

### 2.3. Dating of Change Events

High temporal frequency PlanetScope imagery was used to determine the date on which loss occurred for each parcel where loss was confirmed between February 2016 and January 2020. The removal of Renosterveld within a parcel can take multiple days or weeks to be completed, and even if removal occurs instantaneously multiple days or weeks can elapse before cloud-free image becomes available in which it is possible to confidently confirm that natural vegetation is no longer present. Two dates were therefore assigned to each parcel to describe the temporal nature of Renosterveld loss: the latest date on which it was possible to confirm that intact Renosterveld was still present, and the earliest date on which it was possible to confirm that intact Renosterveld was no longer present. Determining these dates depends on the revisitation frequency of satellite platforms, atmospheric conditions, which determine the image useability, and the nature of land cover change (Figure 2). The near-daily global imaging capability of the Planet satellite constellation and low cloud and haze over this region for much of the year provide sufficient images for these dates to be determined within a few days of each other for many of the Renosterveld loss events detected. The mechanism through which Renosterveld is converted to non-natural land cover, and whether change is continuous or punctuated, influences how discernible land cover change is when examining satellite image time-series'. This in turn influences how accurately loss events can be dated. Given an abrupt disturbance that removes a large amount of aboveground biomass and disturbs the soil surface-such as ploughing-the last date of Renosterveld presence and first date of its absence can be determined within a few days if sufficient imagery is available. Gradual vegetation degradation through continual overgrazing can result in the loss of Renosterveld over a period of multiple years or even decades. This type of land cover change is poorly captured by the approach used here. To assign dates to each parcel of Renosterveld loss

the approximate month and year in which loss occurred was determined using monthly global surface reflectance basemaps available from Planet labs. Once this month was determined, time series' of daily imagery were examined to determine the last date of Renosterveld presence and first date of its absence (Figure 2) using the Planet Explorer online platform. Only those Renosterveld loss parcels in which these two dates could be confidently determined within 10 weeks of each other were used in subsequent exploration and analysis. The rate of loss through time between 2016 and 2020 was examined using the dates assigned and allowed the months and years in which high and low rates of loss occurred to be determined. The size distribution of change events was also explored providing insights into the nature and size of events that contribute most to total loss. Plotting the location, date and size of loss events spatially aided in identifying hotspots of land cover change with the Overberg Region.

Code and data used in this analysis are available at www.github.com/GMoncrieff/renosterveld-data www.github.com/GMoncrieff/renosterveld-change. Due to ongoing investigations location data have been removed to protect the identity of landowners where necessary.

## 3. Results

The random forest classification predicts an area of 3644 ha of potential Renosterveld loss between February 2016 and January 2020. The fitted model achieves an overall accuracy of 90% measured using the spatial k-fold cross validation. For the Renosterveld loss, stable Renosterveld and stable agriculture classes producers accuracy was 72%, 94% and 95%, and users accuracy was 81%, 92% and 91%, respectively. Despite the accuracy of this algorithm, many instances of false positives of Renosterveld loss are observed when manually comparing model output to very high resolution satellite imagery. Fewer instances of false negatives occurred. Common causes of false identification of Renosterveld loss were wildfires within stable Renosterveld (wildfire is a natural occurrence in this ecosystem) and stable agriculture miss-classified as Renosterveld loss. While this is a large overestimate of the actual amount of intact Renosterveld lost, it is desirable to overestimate rather than underestimate the true extent of loss. False positives are removed by the subsequent manual checking, whereas false negatives would go undetected.

After manual removal of false detections and accurate digitisation of boundaries, the final dataset of confirmed Renosterveld loss between February 2016 and January 2020 contained 268 contiguous events covering 478.6 ha. In total, 13% of the area identified as Renosterveld loss by the random forest was confirmed and used in subsequent analysis. While the majority of rejected locations were indeed false positives, others suggest potential change. However, if it was not possible to confirm that Renosterveld removal or degradation resulted in vegetation transitioning from intact to completely lost between February 2016 and January 2020, the site was not included in the final dataset. Therefore, a higher percentage of the area identified by the random forests is indeed Renosterveld loss, and the true area lost is higher than the confirmed 478.6 ha.

The confirmed loss of 478.6 ha represents 0.73% of the remaining 65,684 ha of Renosterveld mapped as either degraded or natural in 2006. It is likely there was far less intact Renosterveld in 2016 and thus the true proportion lost is slightly higher than this figure. Most of the events that result in Renosterveld loss are relatively small, with 57% smaller than 1 ha and 93% smaller than 5 ha (Figure 3). A few large events (18) greater than 5 ha account for 41% of the total area lost.

The date of the 232 (87%) confirmed Renosterveld loss events was determined to within 60 days (the time elapsed between the final confirmed date of Renosterveld presence and first confirmed date of Renosterveld absence)—Figure 4. Of these events the date of a further 134 (50%) could be determined to within 14 days, and 55 (21%) within 7 days. Figure 5 shows the geographic distribution of Renosterveld loss events along with their size and median date of the loss event (the date halfway between the final confirmed date of Renosterveld presence and first confirmed date of Renosterveld absence). In general,

Renosterveld loss occurred later in the western Overberg than the eastern Overberg during this time period. While spread throughout the region, clusters of land cover change are evident in multiple regions (from west to east): northwest of the town of Botrivier, north of the Caledonberg between the town of Caledon and Helderstroom, the Karingmelksrivier west of Napier, southeast of Greyton, east of Protem and both the eastern and western sides of the Breede river north of Malgas.

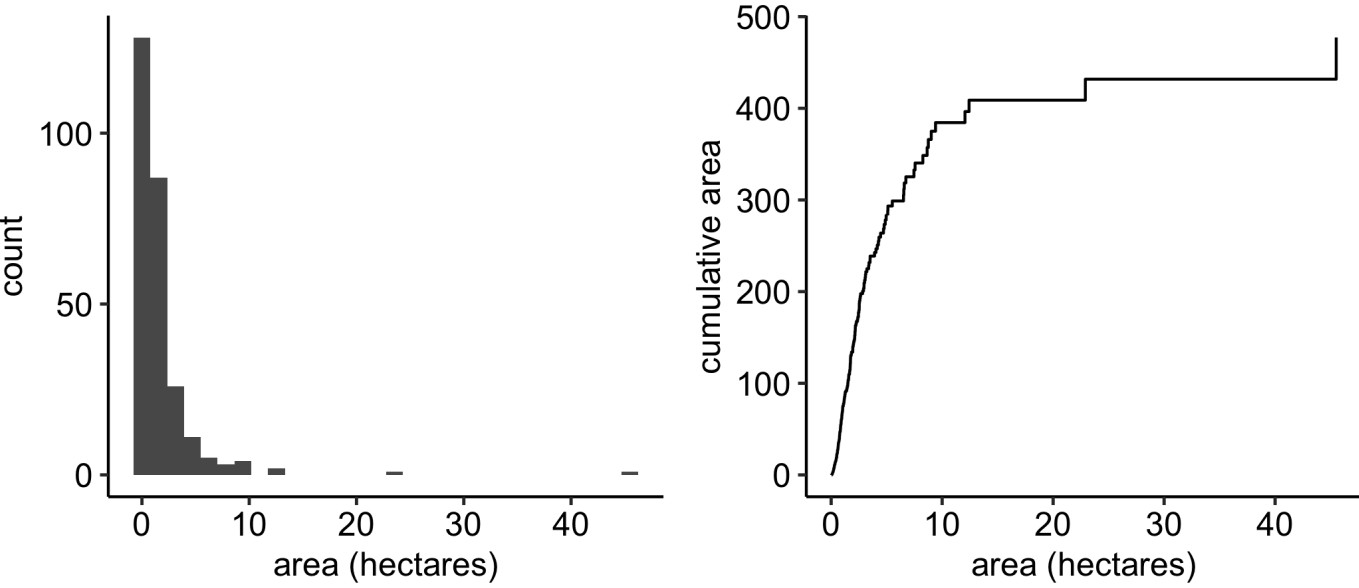

**Figure 3.** Histogram showing the count of Renosterveld loss events of different sizes (**left**), and the cumulative area lost by events of increasing size (**right**).

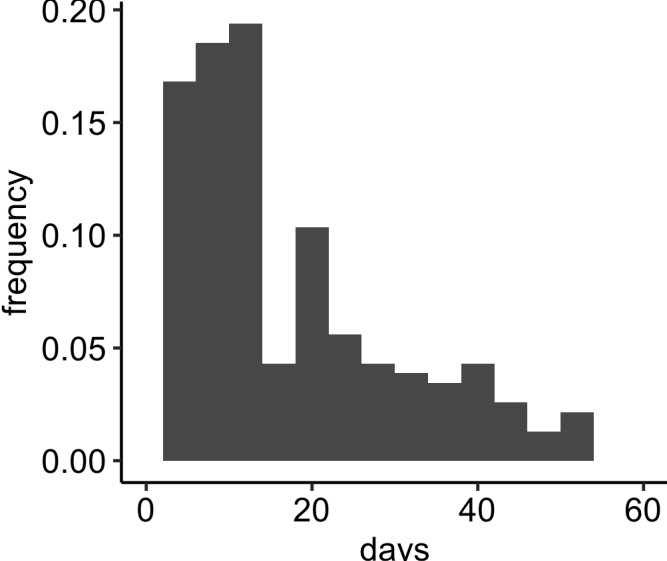

**Figure 4.** Frequency histogram of the time elapsed before a change event could be positively identified for all land cover change events. Time elapsed is calculated as the number of days between the last day of confirmed Renosterveld presence and the first day of confirmed absence.

Renosterveld loss events occurred throughout the year, though across all years the months with the highest concentration of events and the most area lost were February–March, and August–September (Figure 6). Little land cover change took place in the months of December–January and July. Between 2016 and 2020, 2017 saw the most loss

events (despite a large area lost in 2016 due to a few related, large events). No Renosterveld loss was detected and dated in the latter half of 2019.

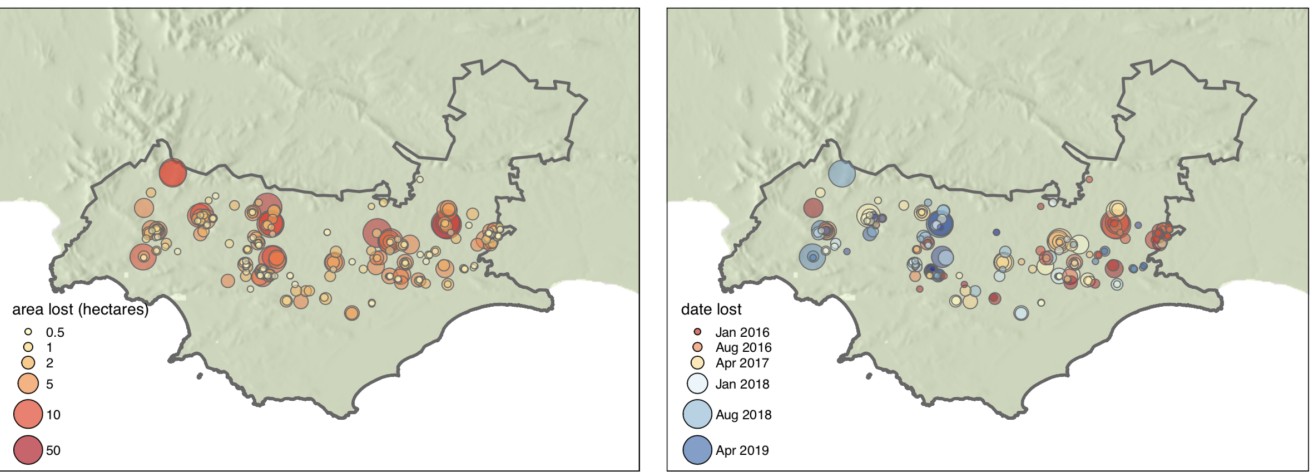

**Figure 5.** The spatial distribution of Renosterveld loss observed in the Overberg district between February 2016 and January 2020. Point size indicates the size of loss events in both panels, whereas colour indicates size (**left**), and the date of loss (**right**).

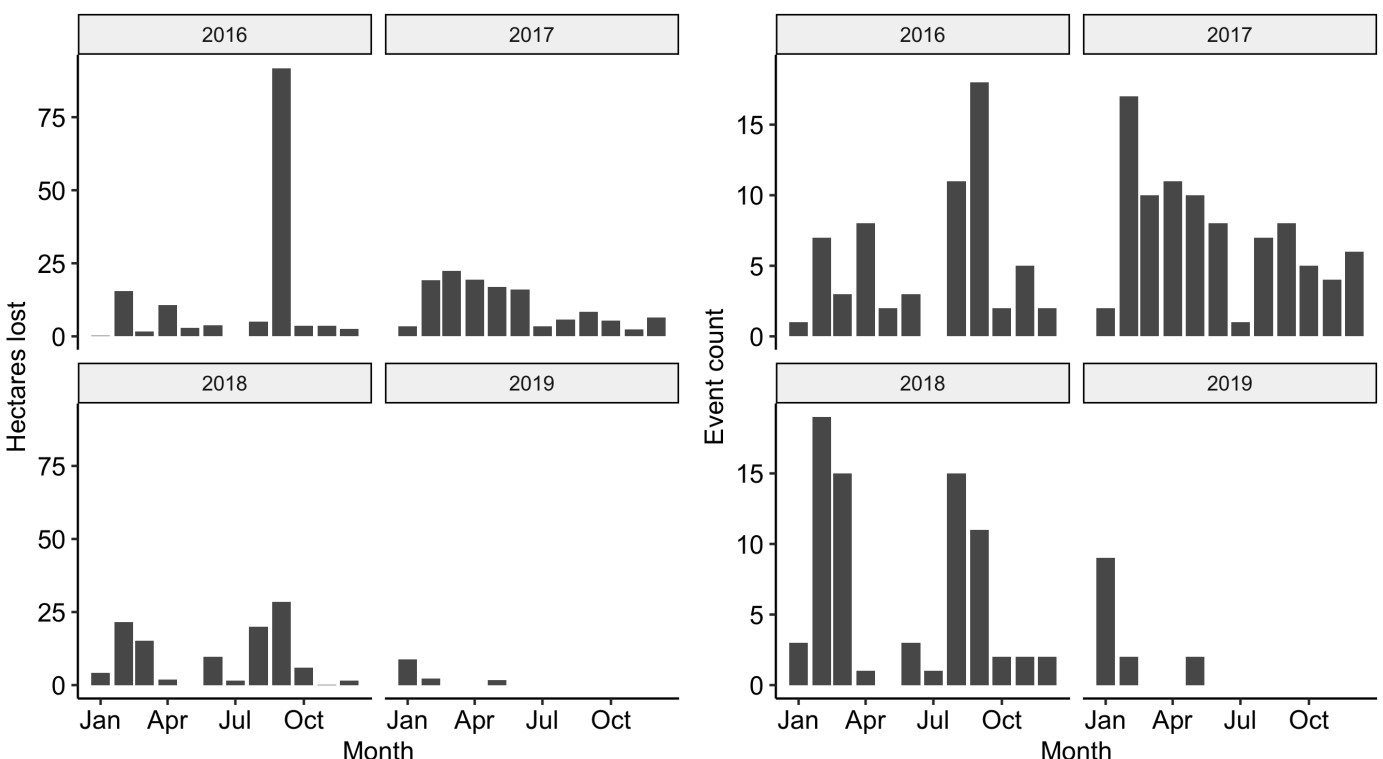

**Figure 6.** Hectares of Renosterveld lost (**left**) and number of loss events (**right**), defined as spatially contiguous polygons, by month and year over the study period.

## 4. Discussion

The vegetation monitored in this study is of exceptionally high conservation value, the density of species threatened with extinction is almost unmatched globally, and every remnant of natural habitat is of enormous value in ensuring the continued survival of the flora and fauna of the Renosterveld [11,19,41,42]. The loss of 478 hectares of intact Renosterveld between February 2016 and January 2020 shows that the current approach to conservation in

the region is failing in its goal of preventing further loss of what remains of the Renosterveld. The ability to monitor the daily progression of land cover change in this region demonstrated here provides information with which to better understand the drivers of Renosterveld loss and the potential for improved enforcement of existing protections. Large inter- and intra-annual variability in rates of loss suggests that actors responsible for land cover change are doing so in response to external factors. Accurate dating of individual events aids in diagnosing these factors and potential future mitigation [43]. For law enforcement and others involved in efforts to prevent further loss, the ability to continuously monitor ongoing change will serve as a strong deterrent, and aid in the deployment of existing resources.

Until recently, frequent monitoring of land cover change at high (<30 m ) spatial resolution has either been impossible or prohibitively expensive. This has precluded the detailed study of land cover change dynamics in highly fragmented and non-forest ecosystems. With the growing time series of data freely available from Sentinel 2 and the unprecedented temporal resolution of the Planet labs Dove constellation it has become possible to reconstruct accurate timelines of land cover change and continuously monitor ongoing change in many of the world ecosystems [22,44]. The approach applied here produced a detailed description of the patterns of land cover change in space and time, allowing 50% of events to be dated within 2 weeks of their occurrence, and the loss of fragments as small as 300 m$^2$ to be detected in a highly fragmented, low-biomass shrubland ecosystem. A significant amount of expert knowledge and manual image interpretation is required to locate and date land cover change events in this ecosystem. However, screening of areas in which change is likely through the rapid creation of a simple training dataset and application of standard classification algorithms greatly improved the efficiency of this task.

This map of Renosterveld change events with associated dates provides a unique dataset with the potential to automate monitoring in the region. Existing approaches to continuous monitoring and detection of change almost exclusively apply an unsupervised approach to detecting land cover change (e.g., [21–24]). Abrupt changes in time-series' of vegetation activity are interpreted as land cover change in this framework. Natural processes that may cause surface reflectance to change, such as natural wildfire or treefall, will be missclassified. Using a dataset in which only land cover change events caused by anthropogenic processes are included to train an algorithm in a supervised manner may separate the signal of unnatural disturbance from that of natural disturbance. The additional knowledge of the date of each event enables training algorithms that will be robust to seasonal changes in vegetation, and hence can be applied continuously as new data become available rather than annually [45,46]. Recent advances in the supervised classification of land cover from annual time-series of Sentinel 2 data provide an excellent starting point for experimentation [47,48]. An alternate approach is to compare remotely sensed vegetation activity to expectations for natural vegetation derived from models accounting for the complex natural dynamics of open ecosystems [27].

All the vegetation types monitored in this study are listed as critically endangered or endangered ecosystems in term of the South Africa's NEMBA legislation. The legislation stipulates that the removal of vegetation in these ecosystems requires prior authorization and a basic environmental assessment. Very few such authorizations were granted in this region over the period studied and therefore it is likely that most of the change reported here can be considered unlawful. Furthermore, it was confirmed that changes affected virgin soil that had remained undisturbed for at least 10 years (but in most cases far longer) as per the CARA legislation. This precludes the invocation of the common defense that old fields rather than intact natural vegetation are being targeted. The information obtained here is being used to aid ongoing investigations and has assisted in the application of administrative penalties for NEMBA contraventions. However, as demonstrated by the rate of change reported here, Renosterveld loss is ongoing. Corrective action to restore lost habitat cannot compensate for the loss of primary natural vegetation to agriculture in this ecosystem [33–35]. The only effective mitigation measures are intervention while habitat

destruction is ongoing or the deterrence of future damage. The implementation of real-time monitoring as described above could allow officials to detect habitat loss soon enough after its initiation that intervention may occur before vegetation removal is completed. Even if this is not possible, reducing the time between the occurrence of an event and expert ground-truthing would aid in the collection of evidence for use in prosecution. The realization that remote sensing technology is in use to monitor threatened ecosystems may itself serve as a deterrent of potential infringements [49].

The nature of Renosterveld loss observed in this study was limited to acute vegetation change within a period of a few days to a few years. Though it is not possible to diagnose exactly the cause of change for each event, the principle factor was ploughing for cultivation. Only a few instance of overgrazing combined with fire were observed to completely transform Renosterveld in this timeframe. The intra-annual patterns of Renosterveld loss reflect patterns of agricultural activity in the region, with the peaks of loss occurring in the months prior to the planting of grains in April and May and before harvesting in October and November. Inter-annual variability may be related to high variability in yields in this region of predominantly dryland agriculture [50]. Poor rainfall over this region over certain years in the study period (2017 and 2019) and macroeconomic factors such as fluctuations in land value may explain some of the patterns seen over the study period, but a longer time series of change would be required to examine these drivers in detail [11,51]. Given that 41% of the area lost occurs through a few incidences of 5 ha and larger, and the concentration of loss within certain sub-regions of the Overberg region, it is likely that a few actors are responsible for much of the loss observed. This may impede interpreting regional patterns of change though time, as influential individuals may respond to idiosyncratic drivers.

The efficiency of the painstaking work of accurately digitizing areas of potential loss and dating individual change events could be improved through the application of improved algorithms and a larger, more diverse training data set [47,48,52]. However, image interpretation will continue to require the expertise of an individual with familiarly of the ecosystem under consideration. Land cover change that occurs slowly over decades or longer as a result of long-term degradation through inappropriate fire regimes, overgrazing or invasive alien species is not captured on the time scale studied here. These factors contribute significantly to the long-term decline of Renosterveld [53]. This may account for the discrepancy between the annual rate of Renosterveld loss reported here of 0.18% per annum and the annual rate reported by [11] of approximately 1%. This limitation may be overcome as longer time series' of data from Sentinel 2 and Planet labs become available, though it will remain impossible to assign a precise date of change to these type of events.

In order to slow and ultimately halt the ongoing loss of Renosterveld a co-ordinated approach is needed. The process described here can provide helpful information to understand the drivers of loss and a lead to the development of tools to monitor change in real-time. This information is already being used by environmental enforcement authorities to aid investigations and deter future infringements. However, progress will be limited unless a systematic approach addressing the underlying causes is developed and behavioural change is incentivised. This has been attempted before [41], and while these efforts have borne fruit, they have not proved sufficient. Efforts to mobilize additional funds and test new approaches to conservation in the region - including using the methods described here - are currently underway.

## 5. Conclusions

Non-forest ecosystems are globally important for the cycling of carbon and water, and are highly biodiverse, yet they are currently poorly monitored using remote sensing. The Overberg Renosterveld is an highly biodiverse shrubland, and has been identified as South Africa's most threatened ecosystem. The rate of loss of 478 ha over 4 years observed here is unsustainable and leading to the extinction of this vegetation type. Accurate mapping and reconstruction of the timeline of events surrounding land cover change in the

region is possible using new remote sensing platforms such as Sentinel 2 and PlanetScope data. These data can be used to aid law enforcement operations, monitor ongoing change and develop plans to mitigate future loss. This demonstrates the possibility of similar data being collected for other shrublands and low tree cover vegetation, and contributing to improved management of threatened ecosystems globally.

**Funding:** This research was supported by the National Research Foundation of South Africa through (Grant No. 118593) as part of the RReTool: Rapid and repeatable tools for monitoring and mitigating global change impacts on natural resources project.

**Acknowledgments:** I am very grateful for the support and encouragement given by Odette Curtis and her passion for conservation in this region. Keletso Moilwe and Lungile Khuzwayo assisted with digitization. Multiple individuals from the Western Cape Department of Environmental Affairs and Development Planning are acknowledged for their input, in particular, those from the Environmental Law Enforcement directorate. Jasper Slingsby provided helpful comments on the manuscript.

**Conflicts of Interest:** The author declares no conflict of interest. The funders had no role in the design of the study; in the collection, analyses, or interpretation of data; in the writing of the manuscript, or in the decision to publish the results.

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
