# Peer review of "Locating and Dating Land Cover Change Events in the Renosterveld, a Critically Endangered Shrubland Ecosystem"

_remotesensing, doi:10.3390/rs13050834_

Round 1

Reviewer 1 Report

I enjoyed reading this manuscript and I provided a minor comments in the attachment

Title: Locating and dating land cover change events in the Renosterveld, a Critically Endangered shrubland ecosystem

General comments:

The manuscript critically examines the land cover change in biodiverse area from 2016 to 2020 in detail. Land cover change is one of the major issues in such area in South Africa. The manuscript is well organized with relatively few typographical errors and a thorough description of the data. It also addresses a highly relevant issue in needs some modification/improvements especially in the following areas:

Title: This is fine.

Abstract: Abstract seems a good. Add few lines why such change took place.

Keywords: Some keywords are repeated with title. Please make sure only keep either in one place.

  1. Introduction: Seems not in flow. You can use some references for the flow of the introduction section.
  • Impact of Land Cover Change on Ecosystem Services in a Tropical Forested Landscape by Sharma et al 2019. Resources.
  • Assessing the financial contribution and carbon emission pattern of provisioning ecosystem services in Siwalik forests in Nepal: Valuation from the perspectives of disaggregated users. Land Use Policy Vol 95.

  1. Data sources and research methods

Why you choose 2016 for the cut-off point year? Please mentioned in the manuscript?

  1. Results:
  • Can you shift your accuracy and other things in methodology section mentioning accuracy and reliability Line No 181-188? Figure 3 is not clear. What is count here (you mean number or others) write more in captions for both figures?
  • Write a short and sweet results. Results are your work and there is no need to add references.
  • In figure 4, you mentioned frequency in points, what does it mean not clear to me?
  • Figure 6 is very interesting and this your total findings. Make a clear description in details. Some data show a very high (e.g. September 2016)

  1. Discussion:

Your discussion section seems a bit lost. Follow the discussion section from the results highlights. You used some literatures in the discussion section. Add some comparison in discussion sections.

  • The manuscript shows a clear high in 2016 and 2017. But it shows a bit low in 2019?
  • If you add the potential reasons for this in the discussion that could be very interesting to the readers.
  • In September 2016 there is very high loss almost 100 hectares? Why this happened? Mentioned in the discussion.

  1. Conclusion: Write a clear and logical conclusion for a broader audience.

Author Response

I am grateful for the helpful comments from the reviewer. I have addressed all the issues raised. Outlined below are my responses to specific issues raised:

Title: This is fine.

Abstract: Abstract seems a good. Add few lines why such change took place.

Keywords: Some keywords are repeated with title. Please make sure only keep either in one place.

A sentence has been added to the abstract highlighting the cause of the land cover change observed. 

Some keywords have been removed

    1. Introduction: Seems not in flow. You can use some references for the flow of the introduction section.

I have added two additional sentences to the introduction to further expand on methodologies for land cover change detection and expanded my description of previous work in this region L64-81

Why you choose 2016 for the cut-off point year? Please mentioned in the manuscript?

2016 was chosen as the starting year for monitoring as it is the first full year for which both Sentinel 2 and PlanetScope data are extensively available. This is indicated on the text (L111-113)

Can you shift your accuracy and other things in methodology section mentioning accuracy and reliability Line No 181-188?

Accuracy results are only reported in the results section and not in methods, so I am not clear to what the reviewer is referring here.

    • Figure 3 is not clear. What is count here (you mean number or others) write more in captions for both figures?
    • Write a short and sweet results. Results are your work and there is no need to add references.
    • In figure 4, you mentioned frequency in points, what does it mean not clear to me?
    • Figure 6 is very interesting and this your total findings. Make a clear description in details. Some data show a very high (e.g. September 2016)

Figure legends have been updated accordingly

    • The manuscript shows a clear high in 2016 and 2017. But it shows a bit low in 2019?
    • If you add the potential reasons for this in the discussion that could be very interesting to the readers.
    • In September 2016 there is very high loss almost 100 hectares? Why this happened? Mentioned in the discussion.

Additional detail on the causes of temporal variation in the rates of land cover changes has been given in the discussion L303-317

    1. Conclusion: Write a clear and logical conclusion for a broader audience.

The conclusion has been targeted at a broader audience by adding a sentence or two with more general context L339-340, L346-348

Reviewer 2 Report

The subject of this paper falls within the scope of Remote Sensing, It is well written and presents the applied methods and obtained results comprehensively. I thus recommend the manuscript for publication but with minor revisions:

General comments

Introduction section: both the study area and its classification are very well described. However, I miss a small review of the methodology used for previous studies in the area from imagery (before Sentinel-2 constellation).

Results section:  Random forest is used for detection, but 87% of these results are discarded. So why wasn't another system used?

Specific comments

Lines 23-25: A reference is needed.

Line 120: Are Sentinel-2 L2A images not used? In areas with a lot of vegetation cover, the highest values of some indices (such as NDVI) using L1C usually give quite different values to L2A.

Figure 1: Although it is explained in the figure caption, it would be much clearer if a legend were included.

Author Response

I am grateful for the helpful comments from the reviewer. I have addressed all the issues raised. Outlined below are my responses to specific issues raised:

I have added two additional sentences to the introduction to further expand on methodologies for land cover change detection and expanded my description of previous work in this region L64-81

I did attempt to use other classification methods (SVM, CART) but these were not as accurate as random forests. I now indicate this in the methods section L145. I also emphasize that I do not intend for the output of the random forest classification to be my final output, and that overprediction of land cover change is preferable to underprediction (L209-212).

I used Sentinel L1C data over L2 data because L2 data is not globally available on Google Earth Engine for the entire historical record, and hence no L2 data is available for our study area at the start date. The choice of images used does however mitigate this, as both images have similar atmospheric conditions, are taken at the same time of year and have clear, cloud free conditions.  This is indicated in the text L137-139

A legend has been added to figure 1